# Phytoremediation Competence of Composite Heavy-Metal-Contaminated Sediments by Intercropping *Myriophyllum spicatum* L. with Two Species of Plants

**DOI:** 10.3390/ijerph20043185

**Published:** 2023-02-11

**Authors:** Yidan Li, Yanyan Song, Jing Zhang, Yingxin Wan

**Affiliations:** College of Biochemical Engineering, Beijing Union University, Beijing 100023, China

**Keywords:** submerged plant, *Myriophyllum spicatum* L., translocation, phytoremediation, sediment, intercropping

## Abstract

A variety of remediation approaches have been applied to reduce the harm and diffusion of heavy metals in aquatic sediments; however, phytoremediation in co-contaminated soils is still not clear. In order to explore the phytoremediation of sediments contaminated by Cu and Pb, two submerged plants with different characteristics, *Vallisneria natans* and *Hydrilla verticillata*, were interplanted with *Myriophyllum spicatum*. By simulating a submerged plant ecological environment, medium-scale-simulated ecological remediation experiments were carried out. The results showed that the two planting patterns were effective in repairing the sediments in the Cu and Pb contaminated sediments. The intercropping of *Myriophyllum spicatum* and *Vallisneria natans* can be used as the plant stabilizer of Cu because of the TF > 1 and BCF < 1, and the intercropping with *Hydrilla verticillata* can regulate the enrichment efficiency of *Myriophyllum spicatum*. The removal rates of Cu and Pb in sediments reached 26.1% and 68.4%, respectively, under the two planting patterns. The risk grade of the restored sediments was RI < 150, indicating a low risk.

## 1. Introduction

De-xing Copper Mine, the largest copper mine in Asia, is located in the Lean River basin of the Poyang Lake. As a large amount of acid water- and wastewater-containing heavy metals are generated during the collection of mineral resources, they eventually flow westward into the Poyang Lake through surface runoff, and cause heavy metal pollution in the sediments [1]. Currently, the technology used for sediment remediation mainly includes hydrodynamic remediation, basement remediation, and phytoremediation technologies [2,3]. Among them, phytoremediation technology is used as a suitable and effective restoration technique. Phytoremediation is a specific class of bioremediation, that uses metabolic processes in plants and the rhizosphere to remove polluting substances from the environment [4]. However, some plants take months or even years to repair [4,5]. In recent years, the use of short-acting crops for restoration has also achieved remarkable results [6]. Therefore, using short-acting crops to enrich heavy metals in ecological restoration needs further research and exploration.

It was found that the specific surface area of submerged plants is much higher than that of floating plants, and all parts of roots, stems, leaves and roots in water environment, can play a role in heavy metal enrichment, so that the effect of heavy metal enrichment is also strong [7]. Because *Hydrilla verticillate (Linn. F.) Royle* can grow rapidly and tolerate and accumulate a number of metal(loid)s, it can be used in phytoremediation [8], *Myriophyllum spicatum* L. has a good enrichment effect on Cu and *Vallisneria spinulosa* could be chosen as a suitable species in ecological restoring of the Pb-contaminated habitats [9,10].

Heavy metal pollution is often complex, because most heavy metals have similar properties. According to research, intercropping of different organisms is more efficient than that of individual species in ecological restoration [11,12,13]. Wang Siqi et al. studied the accumulation of monoculture and intercrop of hyperaccumulator *Solanum nigrum* L. with low accumulation *Welsh onion* conducted to Cd, and found that inter-planting hyperaccumulator with low accumulation crop could normally remediate contaminated soil and obtain an economic benefit. Interspecific cropping of different plants mainly affects the absorption of heavy metal elements in plant root environments. Under heavy metal pollution, plants can secrete organic acids through signal feedback, which can form soluble complexes with heavy metals, to inhibit the transmembrane transport of heavy metals, reduce the bioavailability of heavy metals and reduce their damage to plants [14,15]. At the same time, intercropping can make the full use of light energy, spatial structure and soil fertility.

Heavy metals transfer to sediments due to industrialization in the basin near Le ’anhe Copper Mine in the Poyang Lake. Through the investigation of the sediments near the Poyang Lake Copper Mine, it was found that the risk index of all metals is Cu > Pb > Ni > Zn > Cr > Mn, and the main potential risk factors are Cu and Pb; Cu pollution is more serious [16]. It is of great significance to explore the pollution type and heavy metal pollution degree, and carry out targeted ecological restoration. *Myriophyllum spicatum* L. has a good enrichment effect on heavy metal Cu [17]; however, *Myriophyllum spicatum* cannot fully utilize light and environmental resources in the process of planting alone. The short-term remediation of a plant mixed with different types of submerged plants, in a heavy-metal-complex-contaminated area near copper, needs further study [13,15]. Both, *Vallisneria natans* with developed roots and *Hydrilla verticillata* with rapid growth, reproduce despite root degradation and have a good ecological restoration ability. Thus, it is of great research significance to repair this region after intercropping with *Myriophyllum spicatum*.

In this study, two intercropping modes of *Vallisneria natans* and *Hydrilla verticillata* mixed with *Myriophyllum spicatum* were used to simulate the sediment environment polluted by complex heavy metals. Through short-term cultivation, it was aimed to explore the ability of *Vallisneria natans*, *Hydrilla verticillata*, and *Myriophyllum spicatum* to repair heavy metals after interplanting and compete with each other, and to develop a hybrid remediation method for sediments in this contaminated area.

## 2. Materials and Methods

### 2.1. Source, Collection, and Processing of Samples

According to the actual environment and sampling conditions, the sediments were selected from the middle and lower reaches of Le ’an River basin near Dexing Copper Mine, located in Wannian County, Shangrao City, Jiangxi Province (28°51′26″ N, 116°51′41″ E). During the collection process, 10 sediment samples were removed from the surface, and 1 ton worth of sediment samples were excavated and thoroughly mixed in a ventilated and dry place. Table 1 shows the physical and chemical properties of the sediments. The measurement of physical and chemical indexes refers to the book *Soil Properties Testing, Measurement, and Evaluation Style*. After drying, dead branches and gravel were removed. A sift (50 mm × 50 mm) was used for planting. *Vallisneria natans (Lour.) Hara*, *Myriophyllum spicatum* L., and *Hydrilla verticillata (Linn. f.)* were tested. Plants were purchased from a submerged-plant planting base. Each plant with the same size, number and growth status, was selected as the test plant. The test plant was soaked in 0.2% potassium manganate for 10 min for sterilization, and cleaned with deionized water after soaking.

### 2.2. Planting and Sampling

An uncovered and opaque circular bucket was selected (Height: 1200 mm, Upper diameter: 2480 mm, bottom diameter: 2080 mm), the sifted bottom mud put into the bucket with a height of 20 cm and water added to simulate the state of the bottom mud in the lake. After the water fully permeated the bottom mud in the bucket, *Vallisneria natans (Lour.) Hara*, *Myriophyllum spicatum* L., and *Hydrilla verticillata (Linn. f.)* were weighed, respectively, and the planting was alternated. After planting, the water was filled up, so that the growth state of the plants was similar to the state in the natural lake. The bucket was placed outside and observed for the growth of plants every day. Tap water was added to keep the bucket hydrated. Samples were taken every 12 days, and the test period was 36 days. Plants were randomly selected during sampling, and the plants were cleaned with tap water and distilled water, and dried with filter paper, and the growth parameters of plants were measured. Meanwhile, samples of heavy metals were randomly taken.

### 2.3. Sediment and Plant Metal Determination

#### 2.3.1. Plant Metal Determination

The roots and shoots of each plant were washed with metal free water and rinsed well. Plants were decolorized at 105 °C for 30 min in the oven and dried at 60 °C to constant weight. The dry crop samples were ground, sifted through a 100 mesh. To 0.2 g of treated samples, 2 mL HClO_4_ and 8 mL HNO_3_ was added for digestion. The acids used for the digestion were all guaranteed reagents. The liquid containing the metals was subjected to analysis in inductively coupled plasma atomic emission spectroscopy (ICP-OES 715, Agilen). The multielement standards came from Guobiao (Beijing) Testing & Certification Co., Ltd. (Beijing, China). The ICP was initially calibrated against multielement standards at 0.1, 0.25, 0.5, 1.0 and 2.0 mg/L for Cu, Pb, Ni, Cr, and 0, 1, 2, 5 and 10 mg/L for Mn, Zn. The recovery range was 95.6–107.6%.

#### 2.3.2. Sediment Metal Determination

The sediment was dried at 60 °C in the oven to a constant weight. Soil standard material is the national standard material GBW07427 (GSS-13). After grinding and sifting through a 100 mesh, 0.1 g of the treated sample was weighed and 2 mL of HClO_4_, 8 mL of HNO_3_ and 3 mL of HF were added for digestion. The acids used for the digestion were all guaranteed reagents. The liquid containing the metals was subjected to analysis in inductively coupled plasma atomic emission spectroscopy (ICP-OES 715, Agilen). The multielement standards came from Guobiao (Beijing, China) Testing & Certification Co., Ltd. The ICP was initially calibrated against multielement standards at 0.1, 0.25, 0.5, 1.0 and 2.0 mg/L for Cu, Pb, Ni, Cr, and 0, 1, 2, 5 and 10 mg/L for Mn, Zn. The recovery range was 95.6–107.6%.

### 2.4. The Growth Rate

The growth rate of plants at each stage was calculated by Equation (1)
(1)$$\mathrm{growth}\mathrm{rate}=\frac{m_{2}-m_{1}}{\Delta t}$$
where *m* is the weight of the plant in g and

Δt is the time interval in days.

### 2.5. TF, BCF and BF

The translocation factor (*TF*), bioconcentration factor (*BCF*) and bioaccumulation factor (*BF*) are important in identifying plants suitable for phytostabilization [18], Equations (2)–(4).
(2)$$\mathrm{BCF}=\frac{C\left(roots\right)}{C\left(sediment\right)}$$
(3)$$\mathrm{TF}=\frac{C\left(shoots\right)}{C\left(roots\right)}$$
(4)$$\mathrm{BF}=\frac{C\left(shoots\right)}{C\left(sediment\right)}$$
where *c* is the concentration of the metal in the plant stem and leaves, roots and sediment in mg/kg.

### 2.6. Risk Assessment of Heavy Metals in Sediments

The potential ecological risk index (RI) can measure the sensitivity of a biological community, compared to the overall heavy metal contamination at one site [19]. In this method, the contamination factor of elements, to the potential ecological risk factors and sedimentological toxic response factors, are all covered [20]. Table 2 is the Classification standard of potential ecological risk of heavy metals. The equation for RI is expressed as follows [21], Equations (5)–(7):(5)$$RI=\sum_{i=1}^{n}E_{r}^{i}=\sum_{i=1}^{n}T_{r}^{i}\times C_{f}^{i}$$
(6)$$E_{r}^{i}=T_{r}^{i}\times C_{f}^{i}$$
(7)$$C_{f}^{i}=\frac{C_{i}}{C_{n}^{i}}$$
where the $E_{r}^{i}$ is the potential ecological risk index of the metal, $T_{r}^{i}$ is the biological toxic response factor of the metal i, the $T_{r}^{i}$ values of Pb, Cr, Cu, Zn, Ni and Mn are 5, 2, 5, 1, 5 and 1, respectively. The $C_{f}^{i}$ is the contamination factor of the metal i, which is a ratio between preindustrial records for sediments $C_{n}^{i}$ and present concentration $C_{i}$, the $C_{n}^{i}$ values for Pb, Cr, Cu, Zn, Ni and Mn are 48, 80, 34, 123, 33 and 431, respectively. 

### 2.7. Statistical Analysis

Excel 2021 and SPSS (Version 11.0) were used for data processing and statistical analysis of the experimental results. Heavy metal content of sediments and plants, plant growth (weight) and root growth (weight) were obtained from five parallel data sets. An analysis of variance (ANOVA) was used for analyzing the data of heavy metal content and LSD analysis was used. Differences at *p* < 0.05 were considered significant.

## 3. Results and Discussion

### 3.1. Plant Growth

The two intercropping modes showed a certain tolerance to submerged water toxicity to heavy metals. In the process of culture, compared with the initial growth state of plants, all three plants had a certain growth, the leaves were green and extended, and the length of stems increased. Under the intercropping of *Hydrilla verticillata* and *Myriophyllum spicatum*, the two plants reached the water surface successively and formed a canopy near the water surface, while under the intercropping of *Vallisneria natans* and *Myriophyllum spicatum*, the leaves of *Vallisneria natans* stayed underwater, and the time of W leaves reaching the water surface was longer than that between *Hydrilla verticillata* and *Myriophyllum spicatum*.

The plant tolerance was also reflected in the biomass of the branches and roots under both cropping patterns. Compared with the initial stage, the total biomass of both plants in the planting mode of *Vallisneria natans* and *Myriophyllum spicatum* mixed species reached its highest after 36 days, with an increase of 167% and 24%, and the total biomass of *Myriophyllum spicatum* and *Hydrilla verticillata* in the planting mode of *Hydrilla verticillata* and *Myriophyllum spicatum* mixed seeding reached its highest at 24 days and 36 days of planting, respectively, with an increase of 216% and 20% from the initial stage (Figure 1). 

The plant growth was different under the two intercropping patterns: *Vallisneria natans* only took 12 days to achieve a higher growth rate (Table 3), indicating that *Vallisneria natans* had a strong adaptability to this sediment. By measuring the biomass of a single plant after planting, it was found that the biomass yield was higher on the 24th and 36th days of the two intercropping patterns, among which *Myriophyllum spicatum* had no significant difference between the biomass and the initial state after 12 days of planting in the two modes, the growth rate was the highest and the growth state was better after 24 days of planting, and the plant growth tended to be stable. However, *Myriophyllum spicatum* showed significant differences in growth when intercropped with *Vallisneria natans* and *Hydrilla verticillata*. The growth rate was significantly higher in the intercropping pattern with *Hydrilla verticillata* than in the intercropping pattern with *Vallisneria natans*., indicating that *Vallisneria natans* uses resources in the environment more easily and is more dominant in resource utilization than *Myriophyllum spicatum* [22]. The growth of *Hydrilla verticillata* is rapid and is more dominant in intercropping. 

In addition to the growth rate of plants, root growth and proportion in plants are also important factors in characterizing plant growth [23]. After 12 days of planting, the root biomass of *Vallisneria natans* and *Hydrilla verticillata,* intercropped with *Myriophyllum spicatum,* increased significantly, and the growth of *Myriophyllum spicatum* roots in the two modes showed different characteristics, and the biomass of *Myriophyllum spicatum* roots intercropped with *Vallisneria natans* increased in total biomass. However, the root proportion of *Myriophyllum spicatum* in the intercropping mode with *Hydrilla verticillata* was 0.09 (Table 3), which was not significantly different from the initial root ratio, indicating that the root growth of *Myriophyllum spicatum* was affected in the process of mixing with *Hydrilla verticillata*.

### 3.2. Plant Heavy Metal Enrichment

The experimental crop phytoremediation effect showed that the copper and lead contents of the roots and shoots of plants were increased compared with the initial plant state, among which the Cu content of *Myriophyllum spicatum* root was higher than that of other plants, and the enrichment in the planting mode mixed with *Vallisneria natans* reached a maximum of 425.96 mg/kg after 12 days of cultivation, and the enrichment in the planting mode mixed with *Hydrilla verticillata* reached a maximum of 348.93 mg/kg after 36 days of cultivation (Figure 2). Metal-hyperaccumulating plant species are an interesting example of natural selection and environmental adaptation [24]. The ability of a particular plant to accumulate varies depending on soil and environmental factors. In this study, the concentration of Cu in *Myriophyllum spicatum* is significant: there was no significant difference in the copper enrichment between *Vallisneria natans* and *Hydrilla verticillata* roots mixed with *Myriophyllum spicatum* (*p* > 0.05). However, the highest enrichment of *Vallisneria natans* leaf part was 116.62 mg/kg, which was higher than that of 85.53 mg/kg after 12 days in the *Hydrilla verticillata* leaf culture (Figure 2). The enrichment effect of *Vallisneria natans* and *Myriophyllum spicatum* mixed mode on copper was better, with a total accumulation of 871.69 mg/kg in plants, while the mixed mode of *Hydrilla verticillata* and *Myriophyllum spicatum* had a better enrichment effect on the lead, the accumulation of the lead in plants was the highest, reaching 104 mg/kg (Figure 2).

Since plants are enriched with a certain amount of heavy metals during growth, functional groups, such as carboxyl groups (galacturonic acid in pectin) and hydroxyl groups (hydroxyl groups in cellulose), can be provided because the cell wall of plant cells can be provided [25,26]. They are easy-to-bind metal cations by complexing, coordination, chelation, and ion exchange [27], so that Cu, Pb and other heavy metals are enriched in plants and participate in the metabolic activities of plant cells. Although plants have a certain resistance to heavy metals, their growth rate will be reduced under the action of heavy metal stress to maintain other metabolic activities, and the adsorption efficiency of plants enriched with heavy metals is also reduced at this time. At the same time, because the plant roots have good heavy metal enrichment capacity [28], the proportion of root biomass increases greatly in the initial growth stage, so it has a good enrichment effect on heavy metals, and the proportion of plant roots in the later stage remains stable. Meanwhile the shoots grow rapidly and the rate of heavy metal enrichment decreases at this time.

### 3.3. Heavy Metal Transport in Submerged Plants

The translocation factor (TF), bioconcentration factor (BCF) and bioaccumulation factor (BF) are important in identifying plants suitable for phytostabilization. The ability of plants to translocate heavy metals from roots to shoot is measured by calculating TF and it can indicate the fixation ability and migration efficiency of plant roots for heavy metals [5]. In our study, transport factors for two heavy metals, Cu and Pb, were observed in plants under two planting patterns, TF < 1 (Table 4), suggesting that only a meager translocation of metals took place from roots to shoots. A high accumulation of heavy metals in roots may be ascribed to the complexation of heavy metals with sulphydryl groups, resulting in less translocation of heavy metals to shoots [29]. BCF is the ratio of the concentration of metals in roots to that in soil [30]. The larger the BCF is, the more heavy metals are enriched in plant roots. The difference in the BCF of plants under different planting modes indicates that plant roots are selective to heavy metal enrichment, and metal enrichment is affected by the environment. The BF is the ratio of metal concentration in shoots to the concentration in soil [18]. It can reflect the accumulation ability of heavy metals in sediments by the green part of plants and the transport ability of plants from roots to leaves.

Plants with a TF < 1 and a BCF > 1 are often taken as potential phytostabilizers, because most of the metals get accumulated in their root portion, with very little translocation to the aerial parts [31]. Significant Cu enrichment by *Myriophyllum spicatum* intercropped with *Vallisneria natans* is observed after 12 days of planting, TF < 1 and BCF > 1 (Table 4). After 24 days of culture, the metal enrichment ability of *Myriophyllum spicatum* was decreased because of the damage of heavy metals to the plant roots cells, but the enrichment of roots was still significantly higher than other species. Therefore, *Myriophyllum spicatum* could be considered as a potential phytostabilizer of Cu when intercropped with *Vallisneria natans*. In different stages of the experiment, the TF and BF of *Vallisneria natans* for Cu were significantly higher than that of other species, indicating that *Vallisneria natans* had the strongest transport capacity for Cu and could easily transport Cu from the roots to the leaves, eventually reducting the Cu content in sediment. In the hybrid model of *Hydrilla verticillata* and *Myriophyllum spicatum*, after 12 days of culture, *Hydrilla verticillata* had a large BF and TF, and the results indicated that both the roots and leaves of *Hydrilla verticillata* had an ability to enrich Cu at this stage. The Cu in the *Hydrilla verticillata* could transfer in the roots and shoots. After 24 days of culture, the enrichment rate of Cu by *Hydrilla verticillata* began to decrease. After 36 days, *Myriophyllum spicatum* became the main enrichment plant of Cu.

*Myriophyllum spicatum* behaves differently when intercropped with the two species. *Myriophyllum spicatum* can be used as a potential phytostabilizer for Cu when mixed with *Vallisneria natans*, but has a higher growth rate when mixed with *Hydrilla verticillata*. At 24 days, *Myriophyllum spicatum* mixed with *Hydrilla verticillata* had 2.4 times the growth rate of the other planting patterns. The concentration of Cu in the root of *Myriophyllum spicatum* was only 136.63 mg/kg, due to the main metal enrichment site of root plants (Figure 2). Therefore, there is little effect on the growth of *Myriophyllum spicatum* at this time, and mixed with *Hydrilla verticillata*, it has a higher growth rate. At this time, *Hydrilla verticillata* competition is dominant, as the roots and various parts of it begin to enrich Cu before *Myriophyllum spicatum*. When *Hydrilla verticillata* reached the maximum concentration of Cu, the concentration rate slowed down, and the concentration of Cu was dominant in *Myriophyllum spicatum*. Xue pei-ying et al. also found that *Myriophyllum spicatum* had a good Cu enrichment ability and could be used as a plant stabilizer for Cu [17]. *Hydrilla verticillata* grows and reproduces quickly, giving it an edge over the competition [8]. In the process of planting, intercropping will affect plant growth and plant metabolism [22].

In the case of the metal Pb, although neither plant was a potential stabilizer for Pb, *Myriophyllum spicatum* intercropped with *Hydrilla verticillata* had a larger BCF and a smaller TF, indicating that *Hydrilla verticillata* roots have better enrichment effects on Pb and fix Pb in the roots. Kabata-Pendias et al. have shown that the bulk of the phytoavailable Pb is trapped in the cell wall of the root instead of being translocated in the shoots [32], and mostly accumulates in the rhizodermis and the cortex parts of roots [33]. When *Myriophyllum spicatum* is intercropped with *Vallisneria natans*, the roots of *Vallisneria natans* had higher concentrations and fixations of Pb than *Myriophyllum spicatum*. The enrichment of Pb by *Vallisneria natans* was more dominant.

There was no significant difference in the content of Pb in the shoots of *Myriophyllum spicatum* under the two planting patterns. Moreover, due to the large TF in *Myriophyllum spicatum*, metal Pb was easily transported to other parts of the plant after being absorbed by the roots, and the fixation effect on Pb was poor. However, the concentration of Pb in the roots of *Myriophyllum spicatum* intercropped with *Hydrilla verticillata* was faster than that of *Myriophyllum spicatum* intercropped with *Vallisneria natans*, and the maximum amount of Pb was reached within 12 days. *Myriophyllum spicatum*, interbred with *Vallisneria natans*, takes 36 days.

Similar to Cu, the enrichment of Pb in plants is also related to the competition among plant intercropping and resource utilization [34]. When *Myriophyllum spicatum* was intercropped with *Vallisneria natans*, the concentration of Pb in *Vallisneria natans* was slightly higher than that in *Myriophyllum spicatum*. *Vallisneria natans* also used environmental resources better than *Myriophyllum spicatum*. When *Myriophyllum spicatum* was mixed with *Hydrilla verticillata*, the accumulation rate of Pb in *Myriophyllum spicatum* was significantly higher than that in the other mode, because microbes growing in the biofilms around *Hydrilla verticillata* can have a role in regulating metal uptake and accumulation, through changes in metal bioavailability and speciation [8]. Microorganisms in the biofilm surrounding *Hydrilla verticillata* altered the metal Pb status, increasing the activity of metal ions and promoting the enrichment of Pb in *Myriophyllum spicatum*. A recent study also showed that epiphytic bacteria around *H. verticillate* promoted efficient As (III) oxidation or As (V) reduction in the growth medium, and diverse species of epiphytic bacteria played different roles in As speciation and accumulation in plants [35].

When *Vallisneria natans* is interplanted with *Myriophyllum spicatum*, which can act as a potential phytostabilizer for Cu, *Vallisneria natans* had a better enrichment effect on Pb than *Myriophyllum spicatum*. Because *Vallisneria natans* and *Myriophyllum spicatum* had different photosynthetic sites [36], *Vallisneria natans* is in a different ecological niche and has different utilization of light at different water levels. Moreover, the root system of *Vallisneria natans* is developed and can make full use of resources when intercropped with other species, but the enrichment of heavy metals is time-sensitive. While *Hydrilla verticillata* and *Myriophyllum spicatum* compete for light resources, the two species have different growth cycles: *Hydrilla verticillata* has a short reproductive cycle, rapid growth, rapid reproductive rate after planting, and a better enrichment effect on the metal Pb, and *Hydrilla verticillata* is prone to produce epiphytic bacteria. Microorganisms were able to alter metal availability to assist the enrichment of *Myriophyllum spicatum*, but were more susceptible to heavy metal stress. The reproductive cycle of *Myriophyllum spicatum* was longer than that of *Hydrilla verticillata*, and heavy metal enrichment could still be carried out after the enrichment activity of *Hydrilla verticillata* decreased. *Myriophyllum spicatum* had the ability to enrich Cu and Pb. Therefore, the two mixed species can be applied to the remediation of Cu and Pb composite heavy-metal-contaminated sediment. *Vallisneria natans* and *Myriophyllum spicatum* can be used in areas where Cu pollution is more serious in Cu and Pb combined pollution, and the fixation effect of Cu is good. However, plants enriched with metals should be removed in time, otherwise they will cause secondary release or other ecological hazards. *Hydrilla verticillata* and *Myriophyllum spicatum* can be used for rapid and sustained remediation of Cu- and Pb-contaminated sediment, but require a lot of energy to maintain during plant growth.

### 3.4. Heavy Metal Content and Risk Assessment in Sediments

Figure 3 shows the changes of Cu and Pb contents in sediments with time, under the two planting patterns. Compared with the initial control, the heavy metal contents in sediments decreased significantly. The enrichment of sediments under the two planting patterns also showed heterogeneity, and the Cu removal efficiency reached 26.1% when *Vallisneria natans* was intercropped with *Myriophyllum spicatum*. The removal rate of Pb was as high as 68.4% in the intercropping of *Hydrilla verticillata* and *Myriophyllum spicatum*. Different planting patterns showed different enrichment effects on heavy metals, indicating that the two mixed planting patterns had different effects on plants. With the change of time, the content of heavy metals in sediments fluctuated, and the content of Cu in sediments after 24 days of culture in *Vallisneria natans* intercropped with *Myriophyllum spicatum* showed a slight increasing trend. This indicates that the concentration of heavy metals in plants has a certain threshold, and when the maximum concentration of Cu in plants is exceeded, the plants will start their own protection mechanism and no longer enrich heavy metals in the sediment. Plants can use a wide variety of mechanisms, such as the ability to immobilize metal ions in the roots, root exudates, and the root cell membrane, production of phytochelatins, fixing metals, and displacing either on cell membranes, to protect against heavy metal excess [37]. The sediment also showed the same behavior when *Hydrilla verticillata* was intercropped with *Myriophyllum spicatum*. Pb also showed the same shape after 36 days of cultivation under the two planting modes.

Table 5 shows the change in sediment potential ecological risk index under the two cultivation modes during the cultivation process, which can be used to evaluate the ecological risk of heavy metals in sediments. This study explores the heavy metal polluted area near Dexing Copper Mine in the Le ‘anhe River basin of the Poyang Lake. Therefore, the sediment background value of Nanjishan Basin in the area not polluted by the copper mine is used as the index value. As shown in Table 5, the initial sediment heavy metal elements Zn, Ni, Mn and Cr, all have low ecological risks: Cu presents medium to high risk, and the environmental risk index is 80.04; Pb presents medium risk, and the environmental risk index is 48.69 (Table 5). After submerged phytoremediation, the risk index of the sediment is significantly reduced from moderate pollution to light pollution. Under the two planting patterns, the risk indices of Cu and Pb were significantly lower than the unrepaired level. The risk of Cu reduced to medium from high risk, and the lowest pollution indices were 59.15 and 60.44, respectively. Pb was significantly reduced from medium risk to low risk, and the pollution index was reduced to 18.17 and 15.4 (Table 5). It is not difficult to find that the pollution degree of various heavy metal elements is greatly reduced after cultivation, and the potential ecological risk of sediments is reduced. To a certain extent, these two planting patterns are beneficial to the remediation of heavy metal pollution in sediments.

Based on the Hakanson risk index method, the restoration degree of heavy metals under the two intercropping modes was assessed, and compared with the relevant results before restoration, the reduction of the risk index of sediment indicated that the two planting modes had better restoration effects. Moreover, the different trends of the risk index under the two modes in different periods indicated that the restoration time of different mixed planting was different. To some extent, it can be concluded that different planting methods have certain effects on the release risk of metal elements.

## 4. Conclusions

It was found that the content of heavy metals near the copper mine plant is at risk, exceeding the acceptable limit. After planting submerged phytoremediation, the ecological risks were significantly reduced. At the same time, different planting methods had an effect on the risk of the release of metal elements. *Myriophyllum spicatum* was able to grow in sediment that contained metals when intercropped with *Vallisneria natans* and *Hydrilla verticillata*, and has a certain resistance to metals. After cultivation, the two intercropping patterns had the ability to enrich heavy metals in sediments. *Myriophyllum spicatum* can be used as a plant stabilizer of the metal Cu when intercropped with *Vallisneria natans*, which enriches Cu to the roots, and does not transfer Cu from the roots to the overground part because of the TF > 1 and BCF < 1. Since *Vallisneria natans* and *Myriophyllum spicatum* grow in different ecological niches in the ecosystem, they can make full use of environmental resources, such as light energy. When *Myriophyllum spicatum* intercrops with *Hydrilla verticillata*, *Hydrilla verticillata* can epiphyte a large number of microorganisms to change the bioavailability and morphology of metals, regulate the absorption and accumulation of metals, and regulate the enrichment efficiency of *Myriophyllum spicatum*.

## Figures and Tables

**Figure 1 ijerph-20-03185-f001:**
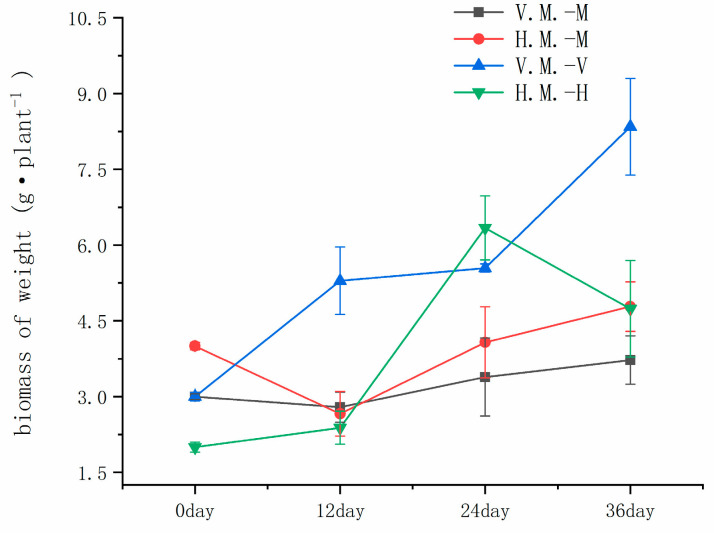
Biomass of plants at different times. The error bars represent standard deviation (n = 5). V. M.: Intercropping *Vallisneria natans (Lour.) Hara* and *Myriophyllum spicatum* L., H.M.: Intercropping *Hydrilla verticillate (Linn. F.) Royle* and *Myriophyllum spicatum* L., V: *Vallisneria natans (Lour.) Hara,* M: *Myriophyllum spicatum* L., H: *Hydrilla verticillate (Linn. f.) Royle*.

**Figure 2 ijerph-20-03185-f002:**
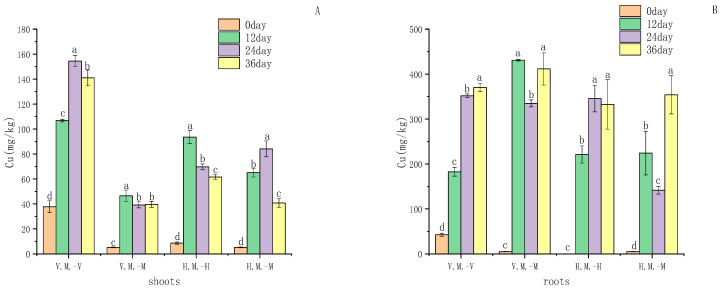
Concentrations of Cu, and Pb (mg·kg^−1^ per plant weight) measured in the roots and shoots of plants *Vallisneria natans (Lour.) Hara*, *Myriophyllum spicatum* L. and *Hydrilla verticillate (Linn. f.) Royle*. The error bars represent standard deviation (n = 3). The letters a, b, c and d denote significant differences between treatments according to the LSD (*p* < 0.05). (**A**): Cu content in plant shoots in mg/kg; (**B**): Cu content in plant roots in mg/kg; (**C**): Pb content in plant shoots in mg/kg; (**D**): Pb content in plant roots in mg/kg.

**Figure 3 ijerph-20-03185-f003:**
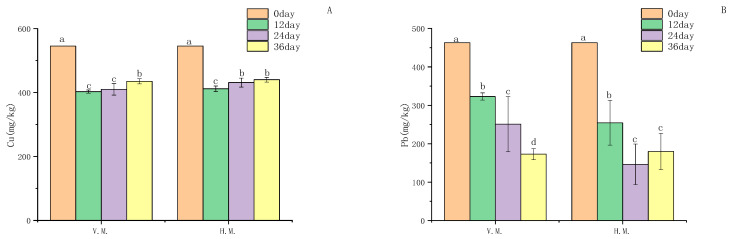
Concentrations of Cu (**A**), and Pb (**B**) in sediments. The error bars represent standard deviation (n = 5). The letters a, b, c and d denote significant differences between treatments, according to the LSD (*p* < 0.05).

**Table 1 ijerph-20-03185-t001:** Physical and chemical properties of sediments.

	pH	OM/(g·kg^−1^)	AN/(mg·kg^−1^)	AP/(mg·kg^−1^)	AK/(mg·kg^−1^)
Sediment	6.12	13.13	116.59	9.16	120.61

Note: AP: available phosphorus, AK: available potassium, AN: available nitrogen and OM: organic matter.

**Table 2 ijerph-20-03185-t002:** Classification standard of the potential ecological risk of heavy metals.

$E_{r}^{i}$ or RI	Potential Ecological Risk
$E_{r}^{i}\;$< 40 or RI < 150	Low
40 ≤$\;E_{r}^{i}\;$≤ 80 or 150 ≤ RI < 300	Moderate
80 ≤$\;E_{r}^{i}\;$≤ 160 or 300 ≤ RI < 600	Considerable
160 ≤$\;E_{r}^{i}\;$≤ 320 or RI < 600	High

**Table 3 ijerph-20-03185-t003:** Plant growth rate and root proportion in different periods.

	Growth Rate				Root Proportion			
	V.M.-M	V.M.-V	H.M.-M	H.M.-H	V.M.-M	V.M.-V	H.M.-M	H.M.-H
0 day	-	-	-	-	0.1055	0.2160	0.1055	0.0483
12 day	−0.0174	0.1912	−0.1115	0.0320	0.1538	0.3071	0.0904	0.0614
24 day	0.0498	0.0209	0.1177	0.3296	0.0806	0.2956	0.0745	0.0495
36 day	0.0279	0.2332	0.0590	−0.1334	0.1083	0.3040	0.0690	0.0476

**Table 4 ijerph-20-03185-t004:** BCF, TF and BF for plants.

		BCF			BF			TF		
		12 d	24 d	36 d	12 d	24 d	36 d	12 d	24 d	36 d
	V.M.-V	0.45	0.86	0.85	0.26	0.38	0.32	0.58	0.44	0.38
Cu	V.M.-M	1.07	0.82	0.95	0.12	0.10	0.09	0.11	0.12	0.10
	H.M.-H	0.54	0.80	0.76	0.23	0.16	0.14	0.42	0.20	0.19
	H.M.-M	0.54	0.33	0.80	0.16	0.20	0.09	0.29	0.59	0.12
	V.M.-V	0.07	0.15	0.18	0.03	0.07	0.10	0.51	0.50	0.57
Pb	V.M.-M	0.04	0.09	0.16	0.04	0.08	0.12	0.97	0.87	0.76
	H.M.-H	0.16	0.46	0.43	0.06	0.10	0.09	0.34	0.22	0.21
	H.M.-M	0.10	0.17	0.12	0.05	0.12	0.11	0.47	0.69	0.92

**Table 5 ijerph-20-03185-t005:** Risk assessment index changes.

		Cu	Pb	Zn	Ni	Mn	Cr	RI
V.M.	0 day	80.0	48.7	7.5	26.1	4.3	5.8	172.5
	12 day	59.2	34.0	4.4	24.4	2.2	3.5	127.6
	24 day	60.2	26.4	4.9	22.3	2.6	4.0	120.4
	36 day	63.9	18.2	5.2	20.2	2.6	4.0	114.1
H.M.	0 day	80.0	48.7	7.5	26.1	4.3	5.8	172.5
	12 day	60.4	26.8	4.6	23.2	2.3	3.7	121.1
	24 day	63.3	15.4	4.7	22.8	2.5	3.9	112.5
	36 day	64.6	18.9	5.2	23.4	2.6	4.4	119.1

## Data Availability

The data presented in this study are available on request from the corresponding author. The data are not publicly available due to the follow-up studies are needed.

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
