# Peer review of "Phytoremediation Competence of Composite Heavy-Metal-Contaminated Sediments by Intercropping Myriophyllum spicatum L. with Two Species of Plants"

_ijerph, 2023, doi:10.3390/ijerph20043185_

Round 1
Reviewer 1 Report
General comments:
The manuscript titled with “Phytoremediation competence of composite heavy metal contaminated sediments by intercropping myriophyllum spicatum L. with two species of plants” study the remediation mechanism and effect of different planting models on composite heavy metal contaminated sediments. The authors have done a great deal of research and the study is instructive for phytoremediation work. However, lots of shortcomings in writing, grammar, logic and theory have been found in this manuscript. The experiment was only conducted for 36 days, so the results are not representative. In short, this article needs serious revision and is not recommended to be published in its current version in the International journal of Environment Research and public health.
Specific comments:
Abstract: The abstract is written in a way that is not quite right. First of all, the order of the abstract methods and results should correspond to each other, and then the conclusions obtained from the results are summarized and their significance is pointed out.
Introduction: the innovation the article and scientific problem are not highlighted in the “Introduction” section, so it is suggested to revise and supplement the content of the preface.
Line 24-37: “Take the phytoremediation as…of land utilization of sludge” This part of the content is not logically strong and the statement is not that clear, please rephrase it.
Line 29-30: “Since ecological…maintaining ecological stability”. This sentence is both redundant and illogical and it is suggested that it be deleted.
Line 35-37: “But some plants take…achieved remarkable results”. This sentence as the end of the first paragraph describes short-lived plants, while the beginning of the second paragraph describes aquatic plants. Please rewrite and emphasize the logic between paragraphs.
Line 45-46: “Since most heavy…heavy metal pollution”. This sentence is meaningless, please rewrite it.
Line 58: Please elaborate why Le 'an River of Poyang Lake is chosen for the study and what is its typicality. You are writing the same way as a report, not a scientific paper used.
Line 92: “organic mater”? or “organic matter” please check other places in MS.
Sediment and plant metal determination: Please add the test method for physical and chemical properties of sediments (i.e., pH, available phosphorus, AK: available potassium, AN: available nitrogen and OM: organic matter)
Statistical analysis: Whether all experimental data met the assumptions of normality and homogeneity for the one-way analysis of variance (ANOVA)? What test was used for significance analysis?
Line 154: Where is the article’s “3. results and discussion”
Line 161-165: “Compared with Myriophyllum spicatum mixed…of the stems increased” Sentences are grammatically incorrect and logically confusing.
Figure: Letters should be placed uniformly above the error bars; The label with the highest average value is a, followed by b, c, d. There are very many and serious problems with the diagrams in the manuscript and the author should understand basic drawing knowledge. Please rework carefully.
Results and discussion: Data sources should be added after each data description (e.g. Figure 1); In this section, please discuss the main results by the data of figures and tables, and focus on scientific problems.
Line 184-190: “However, the growth rate of … is more dominant” Sentences are grammatically incorrect and logically confusing.
Line 205-206: The Latin name of the plants should be italicized, please check other places in MS.
Other points:
There are many serious formatting and grammar problems in the manuscript (e.g., units should be preceded by a blank space), and authors should understand basic knowledge of scientific paper writing. Please rework carefully.
Author Response
Thank you very much for your comments. After careful reading, I found that your comments helped me a lot.
I revised the manuscript after reading, especially about the font, format, spell and other issues. The abstract and introduction have been adjusted, the chart has been remade. The formatting and grammar problems have been modified (line24-37, line29-30, line35-37, line45-46, line92, line154, line161-165, line184-190, ling205-206). And I will continue to check and improve the format and grammar.
About the comment: Please elaborate why Le 'an River of Poyang Lake is chosen for the study? De-xing Copper Mine is located in the Lean River basin of Poyang Lake. As a large amount of acid water and wastewater containing heavy metals are generated during the collection of mineral resources, the result is heavy metal pollution in the Le 'an River. So it is chosen for the study and also reflected in the manuscript.
About the comment: Please add the test method for physical and chemical properties of sediments. The measurement of physical and chemical indexes refer to the book Soil Properties Testing, Measurement, and Evaluation Style. It has been revised and reflected in the manuscript.
About the comment: Whether all experimental data met the assumptions of normality and homogeneity for the one-way analysis of variance (ANOVA)? What test was used for significance analysis? Analysis of variance (ANOVA) was used for analyzing the data of heavy metal content and use LSD analysis.
About the comment: Results and discussion, this part has been modified.
Thanks again for your comments, best regards

Reviewer 2 Report
Thank you for giving me the opportunity to revise the MS entitled “Phytoremediation competence of composite heavy metal contaminated sediments by intercropping Myriophyllum spicatum L. with two species of plants” by Li Yidan and his/her colleagues that was submitted to “IJERPH”. The MS submitted is suitable for IJERPH, and some interesting results were showed. However, there are several requirements that have to consider by the authors. In this regard, the following comments are requested to be addressed by the authors:
Comment 1: Please modify the keyword.
Comment 2: Please carefully check and improve the English.
Comment 3: Line 9 Italic Myriophyllum spicatum. Please check the full text.
Comment 4: Line 41 Italic Hydrilla verticillate. Please check the full text.
Comment 5: Line 48 Italic Solanum nigrum. Please check the full text.
Comment 6: Line 67 Citation should be added here.
Comment 7: Line 85 Pay attention to italics. Please check the full text.
Comment 8: Line 111 Line 116 “0” should be “O”.
Comment 9: Line 155 Insert “3. Results and discussion”.
Comment 10: Will intercropping patterns affect the forms of heavy metals in sediments?
Comments 11: Please carefully check the format of references, such as superscripts, capitalization, journal abbreviations, italic, etc.
For example, Line 431 Line 433 Line 435 Line 437 Line 439 Line 441 Line 451 Line 453 Line 473 Line 491…..
I would suggest that the authors review and include the following recent studies about bioremediation and phytoremediation to improve the manuscript.
1. Su, R.; Ou, Q.; et.al., Comparison of phytoremediation potential of Nerium indicum with inorganic modifier calcium carbonate and organic modifier mushroom residue to lead-zinc tailings. Int. J. Environ. Res. Public Health 2022, 19, (16), 10353.
2. He, L.; Su, R.; et.al., Integration of manganese accumulation, subcellular distribution, chemical forms, and physiological responses to understand manganese tolerance in Macleaya cordata. Environ Sci Pollut R 2022, 29, (26), 39017-39026.
Best regards,
Author Response
Thank you very much for your comments. After careful reading, I found that your comments helped me a lot.
I revised the manuscript after reading, especially about the font, format, superscript and other issues(comment1,2,3,4,5,6,7,8,9,11). I will continue to check and improve the English. I have also read the paper you suggested, which is deeply inspired.
About the Comment 10: Will intercropping patterns affect the forms of heavy metals in sediments? It is a very meaningful question. I also designed related experiments to measure the contents of different forms of heavy metals in sediments. However, this paper mainly discussed the accumulation and migration of total metals by plants under the two intercropping modes, and the analysis of the forms of heavy metals will be analyzed in subsequent papers, so it is not reflected in this paper.
Best regards

Round 2
Reviewer 1 Report
General comments:
The manuscript titled with “Phytoremediation competence of composite heavy metal contaminated sediments by intercropping myriophyllum spicatum L. with two species of plants” study the remediation mechanism and effect of different planting models on composite heavy metal contaminated sediments. The authors have done a great deal of research and the study is instructive for phytoremediation work. However, lots of shortcomings in writing, grammar, logic and theory have been found in this manuscript. The experiment was only conducted for 36 days, so the results are not representative.
Specific comments:
Abstract: The abstract is written in a way that is not quite right. First of all, the order of the abstract methods and results should correspond to each other, and then the conclusions obtained from the results are summarized and their significance is pointed out.
Introduction: the innovation the article and scientific problem are not highlighted in the “Introduction” section, so it is suggested to revise and supplement the content of the preface.
Line 24-37: “Take the phytoremediation as…of land utilization of sludge” This part of the content is not logically strong and the statement is not that clear, please rephrase it.
Line 29-30: “Since ecological…maintaining ecological stability”. This sentence is both redundant and illogical and it is suggested that it be deleted.
Line 35-37: “But some plants take…achieved remarkable results”. This sentence as the end of the first paragraph describes short-lived plants, while the beginning of the second paragraph describes aquatic plants. Please rewrite and emphasize the logic between paragraphs.
Line 45-46: “Since most heavy…heavy metal pollution”. This sentence is meaningless, please rewrite it.
Line 58: Please elaborate why Le 'an River of Poyang Lake is chosen for the study and what is its typicality. You are writing the same way as a report, not a scientific paper used.
Line 92: “organic mater”? or “organic matter” please check other places in MS.
Sediment and plant metal determination: Please add the test method for physical and chemical properties of sediments (i.e., pH, available phosphorus, AK: available potassium, AN: available nitrogen and OM: organic matter)
Statistical analysis: Whether all experimental data met the assumptions of normality and homogeneity for the one-way analysis of variance (ANOVA)? What test was used for significance analysis?
Line 154: Where is the article’s “3. results and discussion”
Line 161-165: “Compared with Myriophyllum spicatum mixed…of the stems increased” Sentences are grammatically incorrect and logically confusing.
Figure: Letters should be placed uniformly above the error bars; The label with the highest average value is a, followed by b, c, d. There are very many and serious problems with the diagrams in the manuscript and the author should understand basic drawing knowledge. Please rework carefully.
Results and discussion: Data sources should be added after each data description (e.g. Figure 1); In this section, please discuss the main results by the data of figures and tables, and focus on scientific problems.
Line 184-190: “However, the growth rate of … is more dominant” Sentences are grammatically incorrect and logically confusing.
Line 205-206: The Latin name of the plants should be italicized, please check other places in MS.
Other points:
There are many serious formatting and grammar problems in the manuscript (e.g., units should be preceded by a blank space), and authors should understand basic knowledge of scientific paper writing. Please rework carefully.
Author Response
Dear reviewer:
Thank you again for taking the time to review and give your comments, I made further revisions to the manuscript. Please see the attachment.
Best regards.

Reviewer 2 Report
The manuscript has been sufficiently improved to warrant publication in IJERPH.
Author Response
Dear reviewer:
Thank you again for taking the time to review and give your comments, I have benefited greatly from them.
Best regards.